# Antarctic Ecosystem Recovery Following Human-Induced Habitat Change: Recolonization of Adélie Penguins (*Pygoscelis adeliae*) at Cape Hallett, Ross Sea

**Jong-U Kim [1], Youmin Kim [1], Younggeun Oh [1,2], Hyun-Cheol Kim [3]**  **and Jeong-Hoon Kim [1,*]**

1. Division of Life Sciences, Korea Polar Research Institute, Incheon 21990, Republic of Korea
2. Department of Life Sciences, Incheon National University, Incheon 22012, Republic of Korea
3. Center of Remote Sensing and GIS, Korea Polar Research Institute, Incheon 21990, Republic of Korea
* Correspondence: jhkim94@kopri.re.kr; Tel.: +82-10-6437-1045

**Abstract:** The human-induced disturbances in Antarctica have caused changes in the structure and function of ecosystems. The Cape Hallett station was established in 1957 and abandoned in 1973. The station was built inside a penguin colony, and during its operation, many penguins were deported. Herein, we compared the number of breeding pairs across different time periods after station decommission and environmental remediation. The station occupied 4.77 ha within the Adélie penguin breeding area, and 349 nests were identified inside the station border in 1960. In 1983, the station's territory decreased to 4.2 ha; meanwhile, 1683 breeding pairs were counted in the old station area. The past station area re-inhabited by Adélie penguins had 6175 nests in 2019. We assumed that recolonization might be particularly related to artificial mounds. The results of the present study confirm the recolonization of Adélie penguins at Cape Hallett for the first time, with visual analysis of spatial-temporal changes. Additionally, we suggest that reconstruction of the artificial or enhanced habitat may be required for successful restoration. Furthermore, continuous species monitoring with specific notes are needed with management interventions to protect Antarctic ecosystems, as well as the Ross Sea region MPA.

**Keywords:** Antarctic seabird; Cape Hallett; human disturbance; Pygoscelis adeliae; recolonization; restoration; Ross Sea region Marine Protected Area

## 1. Introduction

Antarctica is known as a pristine area and is the most remote region on earth. However, Antarctic fauna and flora are affected by climate changes and human activities [1]. The intensity of anthropogenic impact in Antarctica has increased due to international scientific efforts [2]. The human-induced disturbances in Antarctica have caused changes in the structure and function of ecosystems [3]. In particular, the construction and transport activities required to build and maintain stations have become major disturbances to the ecosystem, as well as to Antarctic seabirds [2].

Antarctic seabirds are sensitive and vulnerable to anthropogenic disturbances, even those species with large populations [4], especially land-breeding species, which breed and forage in the terrestrial and marine environments [5]. Results regarding the human impact on Antarctic fauna have continuously been the object of research. For example, cases of negative effects on breeding performance were reported for Gentoo penguins (Pygoscelis papua) [6] and the Southern giant petrel (Macronectes giganteus) [7], according to the reports from tourist expeditions to the Antarctic Peninsula. Moreover, nesting sites of the Adélie penguin (*Pygoscelis adeliae*) were destroyed [4] because of construction activity in East Antarctica and changes in breeding behavior in King penguins (Aptenodytes patagonicus) caused by air traffic in South Georgia [8]. Accordingly, active restoration efforts have been established for some species to address similar conse-

quences [9,10], while the successful restoration is based on proper planning and confirmed by long-term monitoring [11].

Stations built in Antarctica near the penguin colony posed direct threats to the Adélie penguin [2]. The Cape Hallett station is one of these stations, which was a joint scientific base station between the United States and New Zealand, established in 1957 and abandoned in 1973 [12]. It was built within an Adélie penguin breeding colony, and many penguin residents were deported during its operation [13,14]. The Adélie penguin is a colonial seabird breed along the coastline of Antarctica [15]. They have strong natal site return and breed in the same nesting site yearly in the ice-free area close to the ocean [16]. Thus, the nesting site environment is one of the important factors affecting the breeding success of penguins.

The Cape Hallett is an adequate research area for a long-term monitoring program to elucidate the demography of Adélie penguins. The penguin population census has been conducted for decades at Cape Hallett, although there are missing periods. The first survey was done in 1959, with 62,900 breeding pairs counted [13] up to the present, with 43,704 pairs in 2019 [14,17–20] (Figure 1). However, the impacts of the station's presence were undertaken over the decades [2], and initial clean-up, remediation works, and several assessments were conducted for environmental restoration after the station was closed [21–23]. In addition, Wilson et al. [14] summarized the recovery status of Adélie penguins at Hallett station from 1973 to 1988, about three decades ago.

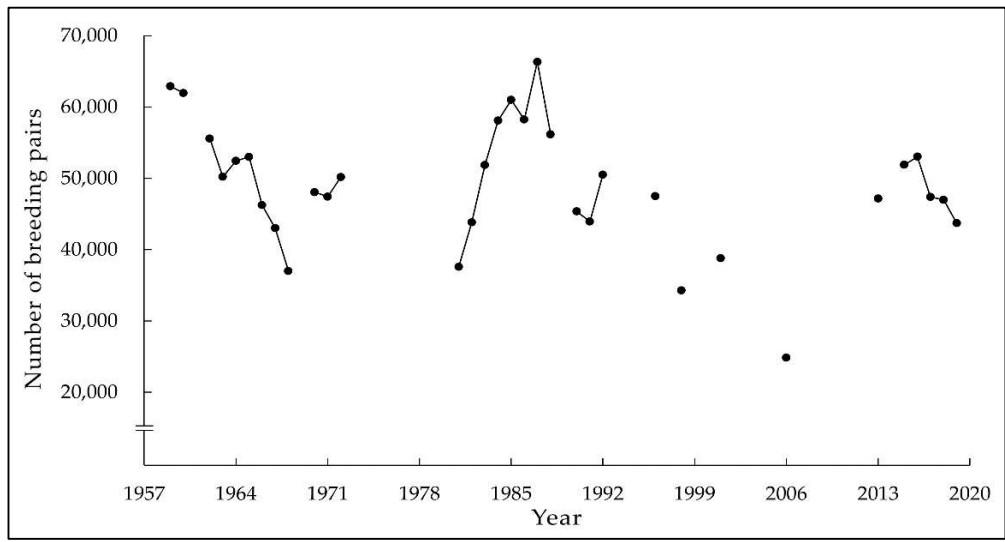

**Figure 1.** Population changes of Adélie penguin breeding pairs at Cape Hallett [14,17–20].

Therefore, an additional survey is required to investigate the recent status of the reoccupation of nest sites by penguins. The aim of this study was to provide verified scientific information about the recolonization of Adélie penguins at Cape Hallett. Furthermore, we would like to present an evaluation of station clean-up efforts and suggest proper habitat restoration plans. We compared the number of breeding pairs and the temporal-spatial changes of nest distribution across different time periods after station decommissioning and environmental remediation.

## 2. Materials and Methods

This study was conducted at Cape Hallett (72°19′ S, 170°13′ E), where about 40,000 pairs of Adélie penguins breed annually during the austral summer (Figure 2). Cape Hallett is located in northern Victoria Land, Ross Sea, the world's largest Marine Protected Area (MPA) [24]. The Ross Sea region MPA was designated to achieve a suite of specific objectives by the Commission for the Conservation of Antarctic Marine Living Resources (CCAMLR), coming into force in 2017 [25,26]. The Research and Monitoring Plan identifies indicator

species for evaluating ecosystem change and, eventually, the effectiveness of the Ross Sea region MPA, as indicated in the CCAMLR Conservation Measure Annex 91-05 [25,27,28]. Moreover, marine predators, including the Adélie penguin, are designated as 'indicator species', also incorporating population monitoring. In addition, Cape Hallett is listed as one of the CCAMLR Ecosystem Monitoring Program (CEMP) sites [29], Antarctic Specially Protected Area (ASPA) no. 106 [30], and Important Bird Area [31].

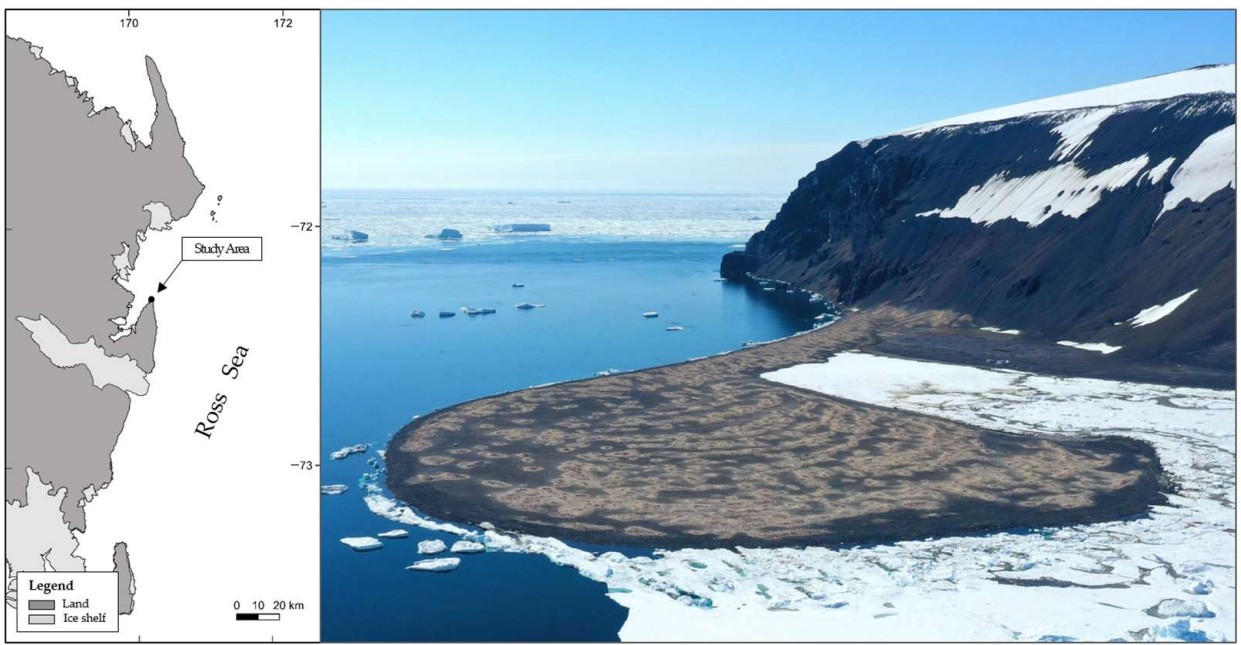

**Figure 2.** Location of Study area (Cape Hallett) in northern Victoria Land, Ross Sea, Antarctica.

We compared the penguin nest distribution inside the Hallett station territory from aerial photographs in chronological order: 1960, while the station was in operation; 1983, after the station was abandoned; 2019, after station decommissioning and environmental clean-up. We used aerial photographs from 1960 and 1983 from the data of the United States Geological Survey (USGS) [32]. The map of the penguin rookery drawn by Reid [33] was referred to for old station area determination (Supplementary Material Figure S1). In addition, aerial photographs of breeding Adélie penguin nests were taken by an unmanned aerial vehicle (UAV, Matrice 600; DJI, Shenzhen, China) with a digital single-lens reflex (DSLR) camera (Eos 5DS; Canon, Tokyo, Japan) in November 2019. The UAV was launched and landed away from penguin breeding colonies, flying 95 m above the terrain [34]. Pix4D (Prilly, Switzerland) software was used to generate a digital elevation model (DEM) and ortho-mosaic image of the breeding colony. We drew the station territory based on the area marked as 'no longer available for nesting' and derived the exact size of the station area.

The penguin colony maps of 1960 [33], 1983, and 2019 were projected to determine the station territory by ArcGIS 10.3 (ESRI, Redlands, CA, USA). We presumed the station territory of 1983 in accordance with nest distribution changes. Moreover, we derived the location of artificial mounds from the images, which were created after the Hallett station clean-up to encourage penguins' breeding [14,21,22]. Then, we counted the number of breeding Adélie penguin pairs inside the station territory in three different periods. Furthermore, we compared the number of nests on the artificial mounds to clarify the recolonization of Adélie penguins.

## 3. Results and Discussion

The Hallett station occupied 4.77 ha within the Adélie penguin breeding area, and 349 nests were identified inside the station border in 1960 (Figure 3). In total, 7580 adults and 3318 chicks were banished from their nesting sites in 1957 during the station construc-

tion [35]. This circumstance occurred because the station site and habitat were preferred by both humans and penguins [1]. The station was operated from 1957 to 1973, which had negative effects on the penguin population in the station area, with the number of breeding pairs declining by up to 59% while the station was in operation [13,14].

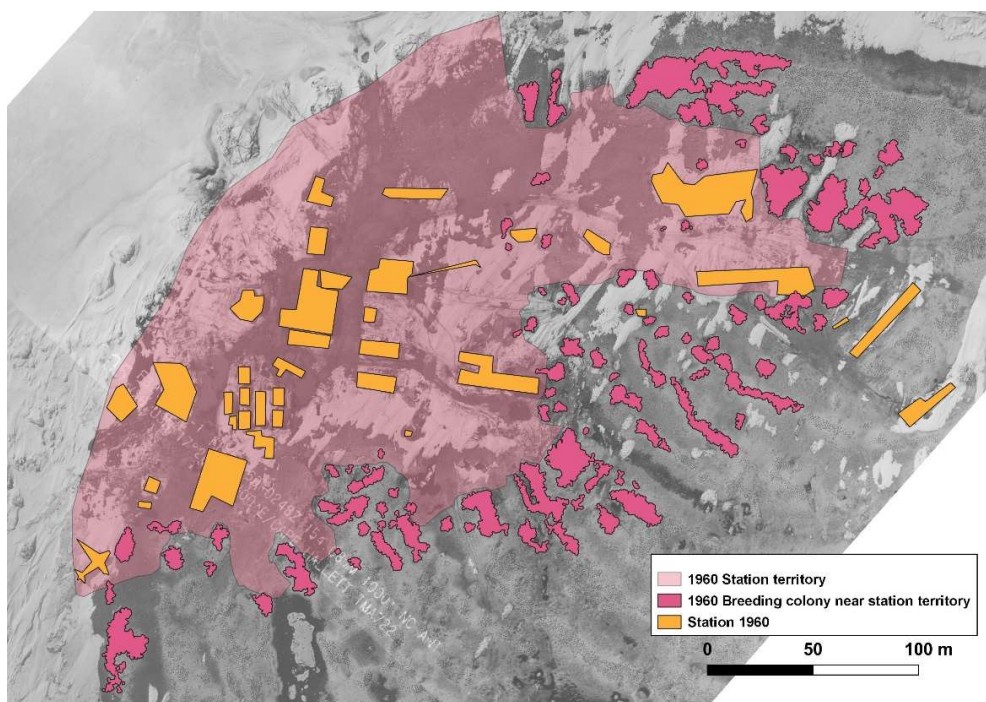

**Figure 3.** Cape Hallett station penguin colony status and the nearby station territory in 1960.

Understanding the impact of anthropogenic activities is necessary [36], but it is also important to know the accurate changes in population for species conservation. In 1983, the station territory decreased to 4.2 ha. Meanwhile, 1683 breeding pairs were counted in the old station area (Figure 4). This might indicate the recovery of the Adélie penguin population, as previously reported [23], although it had been 10 years after the station was abandoned. Initial clean-up of the station was undertaken in 1990, and a number of countries have undertaken the clean-up and remediation of the environment [2,19]. After the implementation of the environmental protocol in 1998, environmental assessment and clean-up efforts were carried out at Cape Hallett from 2000 to 2007 [22,23]. Both American and New Zealand teams implemented remedial actions, cleaned-up contaminated ponds, performed station surface debris removal, building decommissioning, and area restoration by recreating artificial mounds to expedite penguin nesting as a priority [22]. The location of artificial mounds was validated by DEM images (Figure S2), according to descriptions from previous research [13,14].

The past station area was re-inhabited by Adélie penguins, with 6175 nests in 2019 (Figure 5). We could verify the re-occupation of penguins in that a large number of nests expanded into the site previously occupied by the Hallett station. UAV operation was useful in investigating the exact number of breeding pairs with reduced human disturbances [34]. Indeed, the results of our study indicate numerous recolonized breeding pairs compared to the aerial surveys from previous reports [12,14,23]. Successful restoration is possible based on proper planning and long-term commitments [37]. Gilmore [22] informed that penguin nests recovery has been slow in human-modified areas, taking longer than we expected. Forty-six years after station decommissioning, we confirmed that Adélie penguin recolonization has occurred at this site.

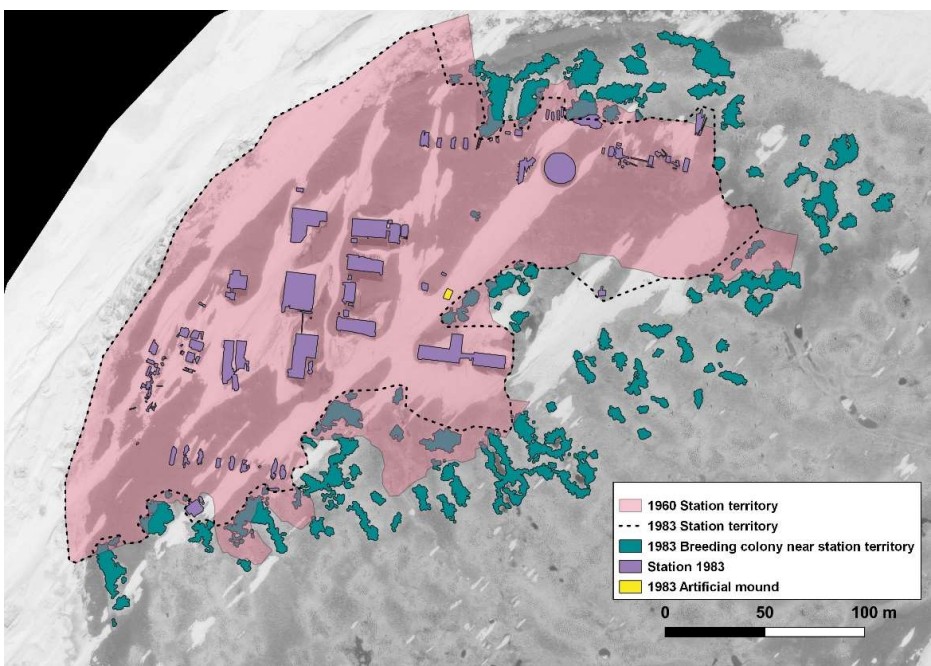

**Figure 4.** Cape Hallett station penguin colony status and the nearby station territory in 1983.

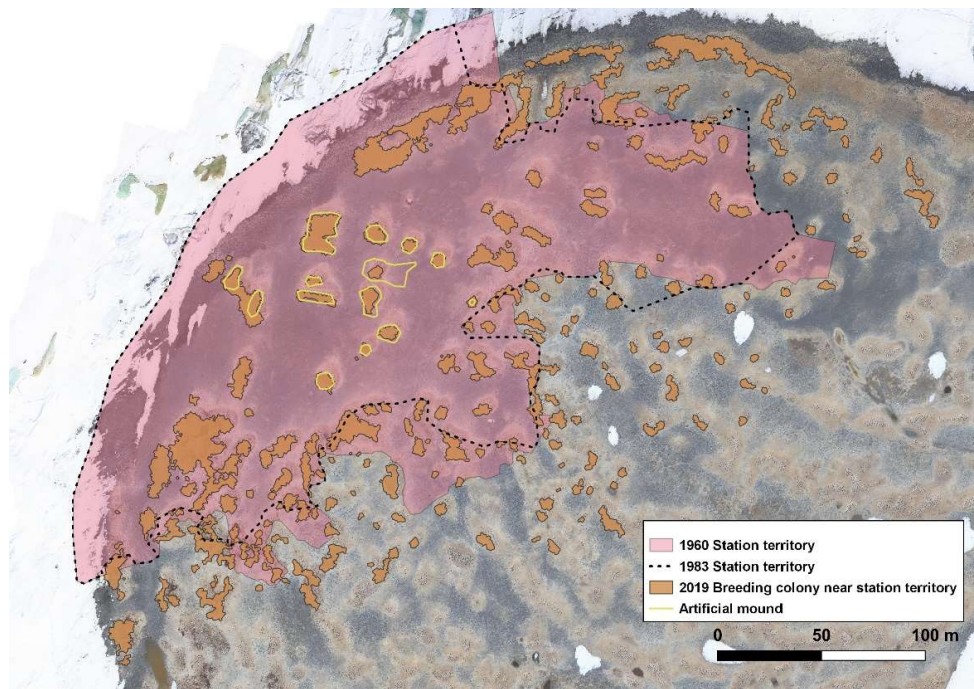

**Figure 5.** Recolonized penguin colony status after the station clean-up project in 2019.

The Adélie penguin population has fluctuated at Cape Hallett (Figure 1). The total number of breeding pairs declined dramatically during the station operation period. Once a decline occurs, it seems to continue even though disturbances are removed [11]. Furthermore, the Adélie penguin population has gone through natural fluctuation for many years and recovered steadily [14]. The population was the highest in 1987, with 66,319 pairs, and the lowest in 2006, with 24,848 counted pairs. Despite the population having decreased and fluctuated, Adélie penguin recolonization is a positive result from a long-term point of view.

To the best of our knowledge, this is the first visually analyzed result of the spatial-temporal changes of penguin colonies in Cape Hallett (Figure 6). In addition, we found that penguins bred on artificially-created mounds and expanded nest distribution around the old station territory. Habitat is a key factor for seabird colonies, and each species has specific nesting environments [38]. Additionally, the available nesting space is one of the underlying components limiting the breeding abundance [39]. Moreover, it helped summarize information about artificial mounds from well-documented pioneering research [14,21,22]. The latest record of artificial mounds used by Adélie penguin pairs was reported in 1988, with 82 pairs [14]. In our study, we observed 1157 breeding pairs nesting on the artificial mounds.

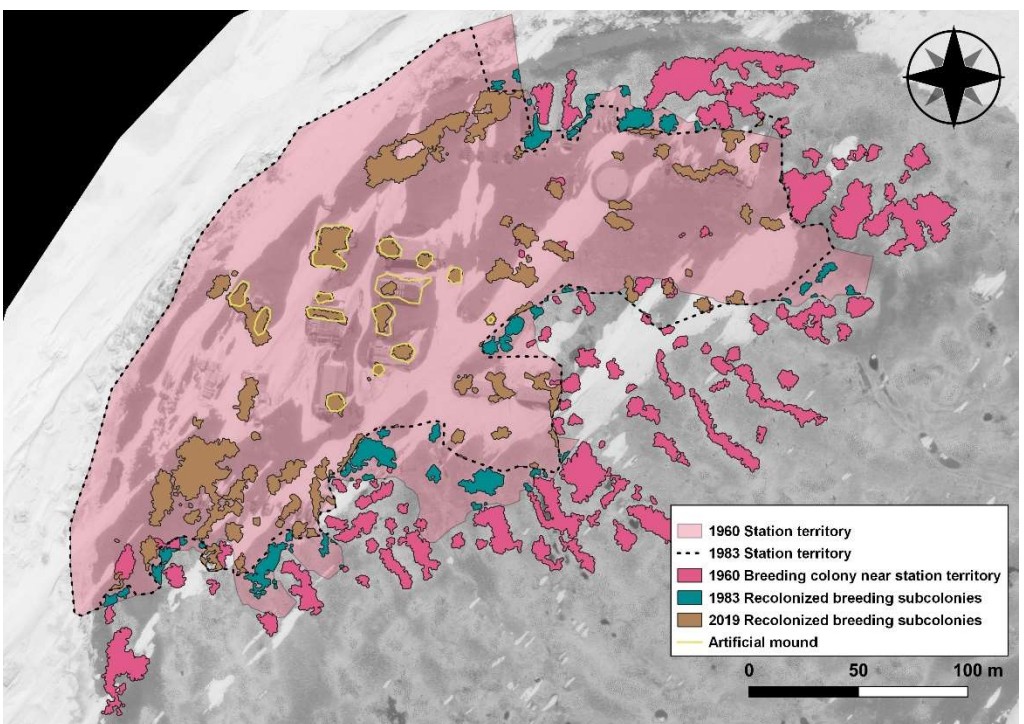

**Figure 6.** Spatial-temporal changes of nest distribution in penguin colonies at Cape Hallett.

We assumed that Adélie penguin recolonization might be related to various aspects of the station clean-up, particularly the artificial mounds. Nesting location is important in determining breeding success in colonial seabirds [6,40]. Seabirds prefer nesting in the central part of the colony because of its ecological benefits, such as making it easier to detect predators and more efficient for communal defense [2,41]. Likewise, the old station was located in the central part of the colony, and the artificial mounds are located in a relatively high position compared to normal ground level. Furthermore, Wilson et al. [14] summarized that penguins showed remarkable adaptability in nesting on artificial mounds, which are well-drained, snow-free, and elevated. In the case of the Cape Hallett penguin colony, the recreated artificial mounds drove penguins to inhabit them with beneficial results.

In addition, detailed historical notes on the station were valuable for a comprehensive understanding of the past and present situation in the Cape Hallett penguin colony. Therefore, continuous ecological studies in Cape Hallett will be necessary, covering various research topics, as well as detailed notes about these studies. Population monitoring should be integrated with the spatial utilization and movement of Adélie penguins and environmental factors [28,36]. These long-term studies coincide with the Research and Monitoring Plan of the Ross Sea region MPA and can provide scientific information on 'indicator species' to contribute to the evaluation of ecosystem changes due to the influence of climate change and anthropogenic disturbances as a CCAMLR member state.

## 4. Concluding Remarks

The Cape Hallett Adélie penguin recolonization is a great example of the restoration of habitat after human disturbances (Figure 7). This study had access to a field survey, well-documented records, and historical reports on the station, environment, and assessment. We suggest that the reconstruction of artificial or enhanced habitats may be required for successful restoration. Additionally, continuous species research and monitoring with specific notes are needed with management interventions to protect the Antarctic ecosystem and the Ross Sea region MPA.

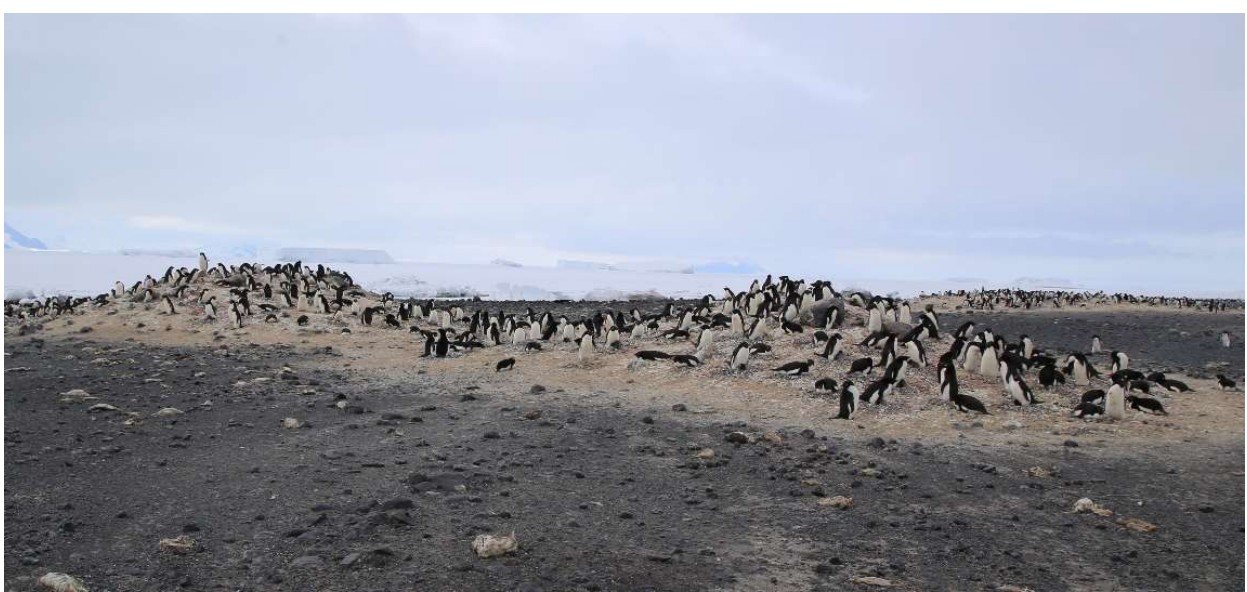

**Figure 7.** Artificial mounds in use by recolonized Adélie penguins at Cape Hallett.

**Supplementary Materials:** The following supporting information can be downloaded from https://www.mdpi.com/article/10.3390/d15010051/s1, Figure S1: The Adelie Penguin Rookery Seabee Spit Cape Hallett Antarctica; Figure S2: Location of artificial mounds at Cape Hallett.

**Author Contributions:** Conceptualization, J.-U.K. and J.-H.K.; methodology, J.-U.K., J.-H.K. and Y.K.; validation, J.-U.K., Y.K., H.-C.K., Y.O. and J.-H.K.; data curation, J.-U.K., Y.K. and Y.O.; writing—original draft preparation, J.-U.K.; writing—review and editing, J.-U.K., Y.K., Y.O., H.-C.K. and J.-H.K.; supervision, J.-H.K.; funding acquisition, J.-H.K. All authors have read and agreed to the published version of the manuscript.

**Funding:** This work was supported by the Korea Institute of Marine Science & Technology Promotion (KIMST) grant funded by the Ministry of Oceans and Fisheries (KIMST 20220547).

**Institutional Review Board Statement:** This study was conducted according to permission from the Korean Ministry of Foreign Affairs in accordance with the Act on Antarctic Activities and Protection of Antarctic Environment. Additionally, this study was carried out in accordance with the 'SCAR Code of Conduct for the use of Animals for Scientific Purposes in Antarctica.'

**Data Availability Statement:** The data are not publicly available due to their usage in the ongoing study.

**Acknowledgments:** We would like to thank Myeongho Seo for his assistance with drone operation and research camp management during our fieldwork. The authors also express thanks for the logistic help from the overwintering members at Jang Bogo Station during the field season. We further acknowledge the anonymous reviewers for their comments which improved the manuscript.

**Conflicts of Interest:** The authors declare no conflict of interest.

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
