# Peer review of "Antarctic Ecosystem Recovery Following Human-Induced Habitat Change: Recolonization of Adélie Penguins (Pygoscelis adeliae) at Cape Hallett, Ross Sea"

_diversity, doi:10.3390/d15010051_

Round 1

Reviewer 1 Report

I enjoyed this paper on the recovery of the breeding population of Adélie penguins in the footprint of the decommissioned Cape Hallett Station. I found the paper well written and suitable for consideration at Diversity. However, I think the paper does require minor revisions prior to publication.

Essentially, the authors should provide more detail in the methods, particularly with respect to how they delineated the station territory, determined what constituted a colony “near” the station territory, and how they derived estimates of penguin abundance within such “near” colonies. Also, the authors would do well to place the changes in abundance that they report in the context of the entire Cape Hallett penguin colony, which seems to have been near 50-60k pairs in the 1960s and is likely higher today, commensurate with the report growth in the subset nesting areas reported here.  I think the inference about the importance of site restoration and other mitigation efforts would be enhanced if the authors can differentiate the general increase in Adélie in the region from the specific re-coloniziation of the former station site.

Several specific suggestions and comments are below.

Abstract – I think it would be helpful to indicate the total size of the Cape Hallett colony through time, not just the numbers of those nesting sites “near” the old station.  This may help differentiate growth in the colony generally from growth in the old station footprint specifically.

Line 12-13 – delete “at Cape Hallett” at the end of the sentence.

Line 40, 42 – here, and in other places, it will be helpful to include details on what and where the human activity caused the negative effects on penguins (tourism in the Antarctic Peninsula in the case of line 40, construction activity in East Antarctica in the case of line 42; air traffic at South Georgia in the case of line 43). Adding the details to the text will be helpful to draw out the idea of how common such impacts are and that they happen all around the continent.

Line 43-44 – the references refer to restoration/mitigation actions in New Zealand, so I suggest a slight revision to say “Active restoration efforts have been established for some species to address similar consequences [9,10].”

Methods  - Generally, there is a need for more detail on several aspects of the analysis. Firstly, how do you determine whether a penguin nesting site is  “near” or “around” the Cape Hallett station territory. And how did you define the Cape Hallett station territory for different time periods? The maps indicate the boundaries, but it is not clear how these boundaries were defined. And please include some details on how estimates of penguin abundance were made based on the aerial photography.  

Line 106-108 – is this decline in breeding pairs just for those nesting sites “near the station” or for the entire Cape Hallett colony? As noted in my general comments, clearly distinguishing the whole colony from the nesting sites “near” the station is a critical gap in the paper.

Line 130 – delete the first “after”.

Figure 2-4 – would it be possible to separately present results for the colony growth inside the 1960s footprint over time – that is, to assess the specific response of the nesting population at the impacted site relative to the entire Cape Hallett? Color coding nesting sites that existed/reappeared within the footprint might help draw attention to that reoccupation, as well. Finally, in Figure 4, consider a hatched marking or another color for the nesting sites on the re-constructed mounds. It’s hard to tell from the figure if some of the artificial mounds are totally covered by nests and where those artificial mounds are.  

Author Response

Thank you for considering our manuscript, and comment about strengths and weakness. We accepted all reviewer's suggestions, and we provide our explanations and responses to the specific questions. We dealt with all comments in attached file. Best regards.

Reviewer 2 Report

This study investigated the recovery of penguin occupation at an abandoned station in Cape Hallett in the Ross Sea region. 3 temporal nest distributions are presented chronologically, which provide useful information on how the ecology recovered overtime. The topic is interesting and the presentation is fine. However, a few issues have to be addressed before it can be considered for publication.

Major:

1) A summary of population change over-time at Cape Hallett as a curve of histogram will be helpful. Maybe the data from MAPPPD can be included with the current study?

2) Give more details to explain why artificial mounds have outstanding adaptability so that penguins prefer them. From your figures, artificial mounds are only a small part of the recolonized area. Do you have data like population density of ordinary former station area and artificial mounds for comparison? If not, it’s not recommended to overly highlight the significance of artificial mounds for the ecological restoration.

3) I think the last two paragraphs of Results and Discussion are out of place, and should be put in the beginning.

4) Is there a standard for the restoration project in this site? A simple assessment would be helpful.

Minor

Line 11: change affected to caused.

Line 32: delete scientific purposes.

Line 33: change affected to caused.

Line 44-45: successful restoration is not based on long-term monitoring, successful restoration can be confirmed by long-term monitoring.

Line 46: Not all stations built in Antarctica are direct threats to the Adélie penguin, only those with improper site choice are.

Line 108: delete breeding pairs.

Line 126: what does numerous mean? I don’t get it.

Line 145: change more easy to easier.

Line 188: delete additionally.

Author Response

(The authors gave the same response as above.)

Round 2

Reviewer 2 Report

All issues addressed.